# Impact of Clinical Pharmacist Consultations on Postoperative Pain in Ambulatory Surgery

**DOI:** 10.3390/ijerph20053967

**Published:** 2023-02-23

**Authors:** Eric Barat, Catherine Chenailler, André Gillibert, Sophie Pouplin, Remi Varin, Vincent Compere

**Affiliations:** 1Department of Pharmacy, CHU Rouen, CEDEX, 76031 Rouen, France; 2Department of Pharmacy, Normandie University, UNICAEN, Inserm U1086, 14000 Caen, France; 3Department of Biostatistics, CHU Rouen, CEDEX, 76031 Rouen, France; 4Department of Rheumatology, CHU Rouen, CEDEX, 76031 Rouen, France; 5Department of Pharmacy, UNIROUEN, Inserm U1234, CHU Rouen, Normandie University, Rouen, CEDEX, 76031 Rouen, France; 6Department of Anesthesiology and Critical Care, CHU Rouen, CEDEX, 76031 Rouen, France

**Keywords:** patient pathway, health care, pharmaceutical care, pharmaceutical interviews, post-operative pain, ambulatory surgery, city-hospital link

## Abstract

Post-operative pain is a common symptom of ambulatory surgery. The objective of this study was to evaluate a pain management protocol integrating a pharmacist consultation. We conducted a quasi-experimental, single center, before-after study. The control group was recruited between 1 March and 31 May 2018 and the intervention group between 1 March and 31 May 2019. Outpatients in the intervention group received a pharmacist consultation, in addition to the usual anesthesiologist and nurse consultations. Pharmacist consultations were conducted in two steps: the first step consisted of general open-ended questions and the second step of a specific and individualized pharmaceutical interview. A total of 125 outpatients were included in each group. There were 17% (95% CI 5 to 27%, *p* = 0.022) fewer patients with moderate to severe pain in the pharmaceutical intervention group compared with the control group, which corresponded to a decrease in the mean pain level of 0.9/10 (95% CI −1.5/10; −0.3/10; *p* = 0.002). The multivariate analysis did not reveal any confounding factors, showing that only the pharmaceutical intervention could explain this result. This study demonstrates a positive impact of pharmacist consultations on postoperative pain in ambulatory surgery.

## 1. Introduction

Pain at home is a frequent symptom of ambulatory surgery [1]. Acute postoperative pain may be explained by a particular physiopathology, however, there is often a lack of anticipation or even absence of medication, in spite of multiple recommendations or expert opinions [2,3]. The consequences of postoperative pain are well-known including chronic pain [4,5], functional impairment, or rehospitalization [6]. Postoperative pain can also lead patients to seek medical care following discharge from hospital [6]. Finally, even with weak opioids, there is a risk of chronic use [7].

Pharmacists could have an impact on the prevention of postoperative pain [8]. According to the literature, the integration of the pharmacist in the care pathway could optimize patient information regarding the correct use of analgesics [9], communication in the multi-professional team [10], community-hospital coordination [11], and patient adherence to the prescribed medications, so that patients become more engaged in their care [12].

In this context, the involvement of a clinical pharmacist in the management of acute postoperative pain in ambulatory surgery seems to be relevant. However, at the present time, no study has been carried out.

The objective of this study was to evaluate a pain management protocol integrating a pharmacist consultation.

The main hypothesis is that adding a pharmacist to the pain management protocol would lower the proportion of outpatients experiencing moderate to severe pain. The secondary hypothesis is that this protocol could improve the overall satisfaction regarding outpatient care.

## 2. Materials and Methods

### 2.1. Ethical Approval

This study (E 2020-87) was approved by the Institutional Review Board (IRB) of a University Hospital in France. Written patient consent was waived by the IRB.

### 2.2. Study Design

We conducted a quasi-experimental, single-center, before-after study. This manuscript complies with the applicable TREND (Transparent Reporting of Evaluations with Nonrandomized Designs) guidelines (Appendix A).

Two groups of outpatients were compared: a control group and an intervention group. Outpatients in the control group were retrospectively included between 1 March and 31 May 2018 from the electronic health records (CDP2^®^, patient files, and Gesbloc^®^, operating room timetable) and outpatients in the intervention group were prospectively included between 1 March and 31 May 2019 by the clinical pharmacist (EB) and two nurse anesthetists.

Outpatients in the control group received the usual preoperative anesthesia evaluation, first with an anesthesiologist and then with a nurse anesthetist. The intervention group received a pharmacist consultation in addition to the usual preoperative anesthesia evaluation. All outpatients, independently of their group, received their consultations several days before ambulatory surgery. Each consultation lasted about fifteen minutes.

### 2.3. Inclusion/Exclusion Criteria

The inclusion criteria were outpatients aged >18 years, who had general or locoregional anesthesia for orthopedic (ORT), odontologic, maxillofacial and ear, nose, and throat (MF/ENT), digestive and visceral, gynecologic, ophthalmologic, plastic and vascular ambulatory surgeries. In the intervention group, outpatients had to have a pharmacist intervention to be included. Exclusion criteria were outpatients who had an ASA (American Society of Anesthesiologists) score ≥3, chronic pain or long-term analgesic treatment, a psychiatric disorder and advanced cognitive impairment, pregnancy, a contraindication to ambulatory surgery, lack of French language proficiency, conversion to conventional hospitalization, a surgical or anesthesia-related complication. Outpatients operated on by surgeons who practiced in 2018 but not in 2019 were not included. Finally, in accordance with the regulations, the outpatients placed under legal protection, guardianship, or curatorship were excluded.

The selection of patients for the control group was carried out retrospectively in the year preceding that of our study. The selection period was identical (year n − 1) to that of the intervention group in order to avoid a seasonal bias. In practice, we identified patients who had an ambulatory surgical procedure between 1 March and 31 May 2018. We applied the inclusion criteria in chronological order and the exclusion criteria patient by patient until the 125 patients were grouped together.

### 2.4. Care Pathway in the Control Group

The organization of the care in ambulatory surgery before the pharmacist intervention is presented in Figure 1:-Several weeks before surgery, the date of surgery was scheduled with the surgeon.-Several days before surgery, the patient attended a 30 min anesthesia consultation. This anesthesia consultation was divided into two 15 min consultations, one with an anesthesiologist and one with a nurse anesthetist.○The anesthesiologist explained to the patient the type of anesthesia, the type of surgery, the pre-operative preparation (taking your medication in the morning, respecting your age, taking a shower), ongoing chronic treatment (in particular anticoagulant and antiplatelet medications), and the analgesic treatment. At the end of this consultation, the patient was prescribed their postoperative analgesic treatment so that they could have them at home before the ambulatory surgery.○The nurse anesthetist reminded the patient of all the logistical modalities inherent to the ambulatory surgery: the time at which they had to stop eating, the rules of hygiene including when and how to take a shower, when and how to organize transport to the hospital, and how to take the chronic treatment in the morning, etc.-Between the anesthesia consultation and surgery, the patient had to go to the local pharmacy to collect their analgesic treatment.-On the day of surgery: after surgery, if the patient had no adverse events, they were discharged home.-The postoperative follow-up was conducted via the SMS platform Memoquest^®^.

### 2.5. Care Pathway in the Intervention Group

As presented in Figure 1, the pharmacist consultation was scheduled during the anesthesia consultation. Consecutively, in the intervention group, the patient had, first, a 15 min consultation with the anesthetist, then, a 15 min consultation with the nurse anesthetist, and finally, a 15 min consultation with the pharmacist, specifically on analgesic treatment based on the prescription written by the anesthesiologist. After the interview with the patient, the hospital pharmacist immediately contacted the patient’s community pharmacist to report the interview. Thus, all post-operative management was entrusted to the patient’s community pharmacist closer to the patient.

### 2.6. Pharmacist Consultation

Pharmacist consultations were carried out by a pharmacy resident. Pharmacist consultations were conducted in two steps. A first step consisting of general open-ended questions to determine the outpatient’s existing knowledge, beliefs, or apprehensions of pain medication (Appendix B). Using this information and the patient’s verbatim carefully recorded by the clinical pharmacist, the consultation was continued with the second step consisting in a more specific and individualized presentation of the prescription and the methods for taking the pain medication, recommendations on the optimal times for taking the pain medication, and finally, a presentation of the drug interactions and possible adverse effects, accompanied by the actions to be taken in the case of adverse effects. At the end of the pharmacist consultation, the patient was given an information sheet summarizing the main points (Appendix C, Appendix D, Appendix E, Appendix F).

Outpatients provided the name of their community pharmacist. Thus, after each consultation, the hospital pharmacist contacted the patient’s community pharmacist by telephone to provide a detailed account of the consultation and to inform them of any difficulties the patient might have.

### 2.7. Follow-Up and Data Collection

The postoperative follow-up was performed via Memoquest^®^ Short Message Service (SMS) response tracking software. An SMS was automatically sent to outpatients at Day (D)1 and D7 after ambulatory surgery. The following items were evaluated: postoperative pain, postoperative nausea and vomiting (PONV), bleeding, fever, or any other event related to the surgery, whether the outpatient had to consult their general practitioner (GP), and whether the outpatient was satisfied with the overall management. Outpatients who did not respond to the SMS or who reported a complication were contacted by telephone by a nurse anesthetist. The following question about pain was asked: “If you had to evaluate your pain on a scale from 0 to 10, with 0 indicating no pain and 10 indicating unbearable pain, how would you evaluate your pain?”.

For overall satisfaction, data were retrieved exclusively from Memoquest^®^. The following question was asked: “How would you rate your satisfaction regarding your care on a scale from 0 to 10, with 0 indicating completely dissatisfied and 10 indicating extremely satisfied”?

In this study, for ethical reasons, we used exactly the same monitoring procedure in both groups. Thus, there was no continuous monitoring for 7 days, or monitoring beyond 7 days, but rather at D1 and D7.

### 2.8. Outcome

The primary outcome was the proportion of outpatients presenting at least one painful episode with an intensity of >3 on a numeric scale from 0 to 10, at D1 or at D7 during the postoperative follow-up. The secondary outcome was the mean overall satisfaction of the outpatients. The numbers of outpatients in each treatment group were balanced with a ratio of 1:1. It was estimated that a pharmacist consultation would reduce the proportion of outpatients with a numeric pain score of >3 from 30% to 15% from a local study [13]. For a statistical power of 80% and a two-sided type I error rate set at 5%, 121 outpatients were required for each group. This was rounded up to 125 outpatients per group, finally, 250 outpatients were recruited. In each group, outpatients were recruited successively using the inclusion/exclusion criteria until the quota of 125 outpatients per group was reached.

### 2.9. Statistical Analyses

All statistical analyses were performed with R software. The significance level used was 5%. No interim analysis of the primary endpoint was performed.

The primary analysis was performed per protocol. Outpatients were allocated to the intervention or control group according to whether they were included in the retrospective or prospective period, and, in the prospective period, that they had received the intervention. Outpatients converted to conventional hospitalization were excluded.

The proportion of outpatients with a pain score of >3 on the numeric pain scale at D1 or D7 was estimated in each group and compared between groups by Wald’s test with a 5% type I error rate. The two-sided 95% confidence interval (CI) of the absolute risk difference was estimated by Wald’s method. The mean pain levels were also obtained in each group and compared using a Student test with a 95% confidence interval.

A sensitivity analysis was performed excluding outpatients with missing data in both groups.

A multivariable general linear model was estimated to assess the absolute risk difference of pain in the intervention group with adjustment for pre-prescribed World Health Organization (WHO) analgesic level (1 to 3) [14], type of anesthesia (general or locoregional), and surgical specialty (orthopedics, plastic surgery, etc.).

Satisfaction levels were compared between the two groups by Student’s *t*-tests at the 5% significance level and the accompanying 95% two-sided CIs of the difference between groups were reported. For this analysis, patients with missing data on satisfaction were excluded. No multiple testing procedures were performed.

## 3. Results

### 3.1. Flowchart

Recruitment and monitoring of each group was carried out as shown in Figure 2. The mean (±CI) of the inclusion rate in each group was 8.93 (±5.82) inclusions/week for the control group and 7.75 (±4.01) inclusions/week for the intervention group.

### 3.2. General Characteristics of Outpatients

The study compared two groups of outpatients over two different time periods. A total of 125 outpatients were included in each group. Outpatients in the control group were retrospectively included between 1 March and 31 May 2018 and outpatients in the intervention group were prospectively included between 1 March and 31 May 2019.

The baseline characteristics of the outpatients are presented in Table 1.

The mean age was 44.07 years (SD 17.38) in the control group and 47.28 years (SD 18.33) in the intervention group (*p* = 0.16); the male/female ratio was 1.15/1.31 without a significant difference between groups. Two types of anesthesia were used: general (GA) and locoregional (LRA) anesthesia (Table 1). A total of 80 (64%) outpatients in the control group and 89 (71%) in the intervention group had GA without a significant difference between groups. A systematic or on-demand WHO step 2 analgesic, with or without a step 1 analgesic, was prescribed to 98 (78%) outpatients in the control group and to 96 (77%) outpatients in the intervention group without a significant difference between groups. The drugs prescribed during the study by the anesthesiologists were paracetamol (step 1), ketoprofen (step 1), tramadol (step 2), nefopam (step 1), codeine (step 2), and morphine (step 3).

A significant difference was observed between the control group and the intervention group for surgical specialty (*p* = 0.0007, see Table 1): 33% and 46% for orthopedics, 24% and 10% for MF/ENT, and 28% and 18% for plastic surgery, respectively.

### 3.3. Pain

In the control group and the intervention group, 96 (77%) outpatients and 111 (89%) outpatients, respectively, had complete data for pain assessment (*p* = 0.45). According to Wald’s test (primary analysis) without adjustment, the difference was estimated at −17% (95% CI −5 to −27%, *p* = 0.022) for the group with pharmaceutical care. In the control group, the average pain was 2.6/10 and in the interventional group, the average was 1.7/10. The difference in pain between the two groups was significant according to the Student’s t test with −0.9/10 (95% CI −1.5/10; −0.3/10; *p* = 0.002). In the analysis with adjustment on the prescribed analgesic WHO step (1, 2 or 3), anesthesia (GA or LRA), and surgical specialty (odontology, MF/ENT, plastic, other), and pharmaceutical care, the difference of risk of pain in the intervention group was estimated at 16% (95% CI 4 to 27%, *p* = 0.03). The multivariable analysis with effects of other variables is shown in Table 2.

A sensitivity analysis was conducted with a pain threshold set at >5 or >7 without statistical adjustments and there were significant differences between the control and intervention groups (Table 3).

### 3.4. Satisfaction

Data on satisfaction, on a scale from 0 to 10, were missing in 40 (32%) outpatients in the control group and in 15 (12%) in the intervention group. In a complete case analysis, the mean (standard deviation) satisfaction was estimated at 8.77/10 (1.96) in the control group and at 8.97/10 (1.44) in the intervention group. Using the Student’s t-test, the difference was not significant between the two groups (*p* = 0.41). The difference in satisfaction between the groups was estimated at −0.20/10 (95% CI −0.28/10 to +0.68/10).

## 4. Discussion

This study confirmed the main hypothesis that adding a clinical pharmacist to a pain management protocol could lower the proportion of outpatients experiencing moderate to severe pain. Indeed, even with the pessimistic maximum bias hypothesis, there were 17% fewer patients with moderate to severe pain in the group that had a pharmacist consultation, which corresponded to a decrease in the mean pain level of 0.9/10.

The effect of a pharmacist consultation on pain > 5 was significant, suggesting a benefit for patients with more severe pain. The intensity of postoperative pain is a primary cause of delayed discharge or readmission [15], and a decrease in severe pain could represent a benefit. We did not find any significant benefit of the pharmacist consultation on satisfaction, despite published studies on the contribution of pharmacist consultations to patient satisfaction [10,16]. The fact that the level of satisfaction was already high in the before period, with little room for improvement, and that many factors unrelated to the pharmacist may impact satisfaction such as nurse, anesthesiologist, and surgeon behaviors and the health status of the patient, may explain why the difference was not significant.

The proportion of outpatients with postoperative pain in the control group (31%) was consistent with the data in the literature, with an overall incidence of around 30% reported in different European [17,18] or North American [19,20] studies.

The observed decrease in pain in the group that had a pharmacist consultation was probably due to the fact that analgesic treatment was planned (i.e., taking analgesic medication at prescribed times rather than waiting for the onset of pain). Indeed, it has been shown that pharmacist consultations promote adherence to treatment [21]. Moreover, Robaux et al. reported better pain management when postoperative analgesia was planned beforehand. As in our study, these authors also found the same decrease in pain [22].

Our pharmacist consultation helped to strengthen the community–hospital link. Indeed, in the intervention group, the community pharmacist systematically received a call from the hospital pharmacist. We suggest that this community–hospital link may partly explain the decrease in pain in the intervention group. Indeed, it can be hypothesized that this transmission of information to the community pharmacist made it possible to renew the information to the patient, thus reinforcing its impact, especially regarding adherence to treatment, while allowing the hospital pharmacist to be particularly vigilant with regard to postoperative follow-up.

In the literature, a clinical pharmacist consultation coordinated within a multidisciplinary team led to similar improvements in chronic pain and acute postoperative pain. Indeed, Slipp et al. [16] showed that this organization significantly decreased pain and the duration of disability compared to an intervention without a pharmacist, and increased the patient satisfaction compared to an intervention with physicians only. We observed an improvement in acute postoperative pain, but the short follow-up in our study did not allow us to assess chronic pain.

This study had several limitations. Methodologically, the two groups were compared during two different time periods (i.e., two different years). A possible seasonal or surgeon effect was reduced by including only those procedures that were performed by surgeons practicing at our center during the same months in the two different years. However, it is possible that the types of surgery changed between the two time periods, with potential unmeasured confounding factors. Since the surgical specialty adjustment did not change the effect estimate, this may not be a major confounding factor. The large inclusion criteria may lead to a risk of heterogeneity (types of surgery and anesthesia). However, these criteria were chosen to obtain a representative sample of outpatients. The retrospective nature of the study and the use of SMS software for patient follow-up limited the number of variables that could be considered such as adherence to treatment, continuing analgesic treatment or not, or the possible progression to chronic pain. However, in the literature, the transformation of acute pain into chronic pain is well-described, and the presence or absence of post-surgical pain plays an important role [5]. Thus, the results of this study suggest that pharmaceutical care may contribute to a decrease in chronic postoperative pain, especially since, in addition to a decrease in postoperative pain, the pharmacist’s involvement allowed for multimodal pain management, as recommended in the literature [23]. Moreover, automated monitoring at day 1 and at day 7 did not allow for a reliable assessment of pain between these times. Similarly, the use of this software led to a greater number of missing data in the control group than in the intervention group, with a risk of attrition bias and differential measurement bias with a telephone call compared to an SMS. These different risks seem to be controlled because the main result remained significantly in favor of the pharmacist consultation, even in the maximal bias hypothesis.

## 5. Conclusions

This study is part of an ongoing protocol for the management of postoperative pain. Pain is multimodal and pain management requires coordinated and complementary actions involving a multidisciplinary team of health professionals. This study showed that adding a pharmacist to an existing pain management team could contribute to the overall effort to improve postoperative pain management. In practice, a pharmacist consultation could improve pain management by freeing up physician and nurse time. The encouraging results of this study, focusing on the contribution of pharmacists, should not overshadow the work of other team members as anesthesiologists, surgeons, nurses, and all other health professionals in the hospital or community involved in pain management. On the contrary, the aim of this study was to highlight the relevance of a pharmacist consultation in the overall management of postoperative pain and to promote the broadest possible multiprofessional collaboration. Now, a cost-effectiveness evaluation is needed before the widespread deployment of pharmacist consultations.

## Figures and Tables

**Figure 1 ijerph-20-03967-f001:**
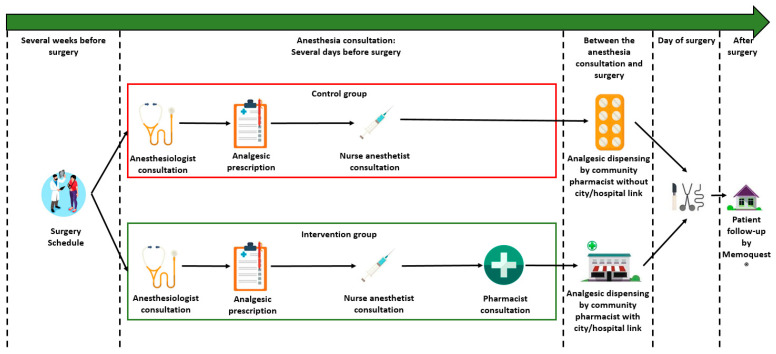
Care pathway in the control group and intervention group.

**Figure 2 ijerph-20-03967-f002:**
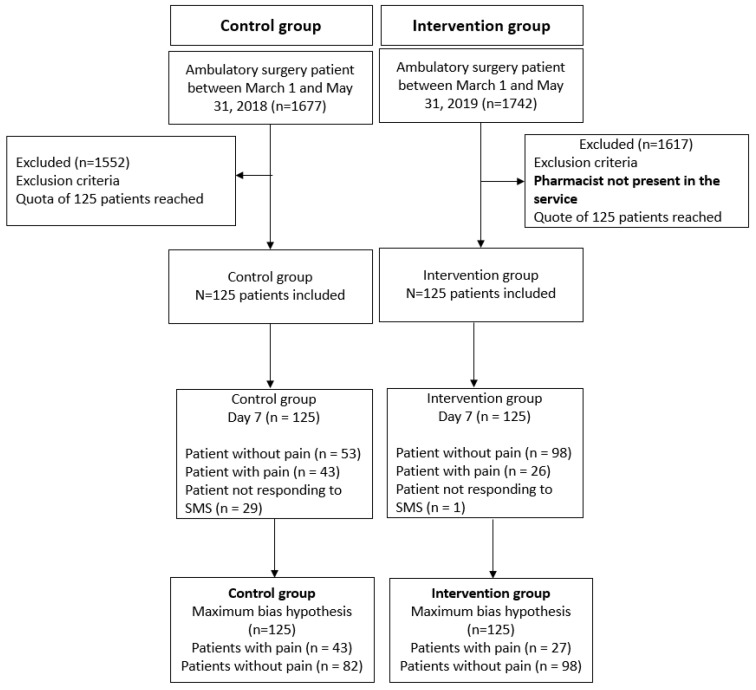
Flowchart.

**Table 1 ijerph-20-03967-t001:** Baseline characteristics of the outpatients compared between the control and intervention groups.

Data	Control Group *n* (%)	Intervention Group *n* (%)	Overall *n* (%)	*p*-Value
Age, mean (SD ^1^)	44.07 (17.38)	47.28 (18.33)	45.68 (17.9)	0.16
Anesthesia				0.28
Locoregional	45 (36%)	36 (29%)	81 (32.4%)	
General	80 (64%)	89 (71%)	169 (67.6%)	
WHO ^2^ analgesic step				0.76
1	26 (21%)	29 (23%)	55 (22.0%)	
2	98 (78%)	96 (77%)	194 (77.6%)	
3	1 (1%)	0 (0%)	1 (0.4%)	
Surgical specialty				0.0007
Orthopedics	41 (33%)	58 (46%)	99 (39.6%)	
Plastic surgery	35 (28%)	23 (18%)	58 (23.2%)	
MF/ENT ^3^	30 (24%)	12 (10%)	42 (16.8%)	
Odontology	5 (4%)	16 (13%)	21 (8.4%)	
Visceral/Digestive	7 (6%)	10 (8%)	17 (6.8%)	
Vascular	2 (2%)	5 (4%)	7 (2.8%)	
Gynecology	3 (2%)	1 (1%)	4 (1.6%)	
Ophthalmology	2 (2%)	0 (0%)	2 (0.8%)	

^1^ SD: standard deviation; ^2^ WHO: World Health Organization; ^3^ MF/ENT: maxillofacial/ear–nose–throat.

**Table 2 ijerph-20-03967-t002:** Multivariable general linear model explaining the risk of pain intensity >3.

Data	Absolute Difference of Risk of Pain [95% CI]	*p*-Value
Group		
Control	0 (reference)	
Intervention	−16% [−27 to −4%]	0.02
WHO ^1^ step (linear effect)	+5%/step [−8 to 19%]	0.43
Anesthesia		
Locoregional	0 (reference)	
General	+1% [−13 to 15%]	0.85
Surgical specialty		
Orthopedics	0 (reference)	
Plastic surgery	−2% [−17 to 13 %]	0.79
MF/ENT ^2^	+9% [−8.8 to 27.5%]	0.31
Odontology	+6% [−16 to 29 %]	0.59
Other	+4% [−16 to 24%]	0.72

^1^ WHO: World Health Organization; ^2^ MF/ENT: maxillofacial/ear–nose–throat.

**Table 3 ijerph-20-03967-t003:** Comparison of pain at different thresholds between the control and intervention groups with two different hypotheses about the missing data.

Pain Intensity ^1^	Control Group *n* (%)	Intervention Group *n* (%)	Unadjusted Difference [95% CI]	*p*-Value
Pain intensity > 3	30 (31%)	16 (14%)	−17% [−27 to −5%]	0.022
Pain intensity > 5	14 (15%)	5 (5%)	−10% [−18 to −1%]	0.03
Pain intensity > 7	2 (2%)	0 (0%)	−2% [−7 to 3%]	0.22

^1^ Pain intensity measured using a numerical scale from 0 to 10; where 0 is no pain; 1 to 3 corresponds to mild pain; 4 to 6 corresponds to moderate pain; 7 to 10 corresponds to severe pain.

## Data Availability

Not applicable.

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
