# Peer review of "Impact of Clinical Pharmacist Consultations on Postoperative Pain in Ambulatory Surgery"

_ijerph, 2023, doi:10.3390/ijerph20053967_

Round 1

Reviewer 1 Report

1.       What is the rationale behind excluding nurse consultations from this study? 

2.       What is the rationale behind excluding Psychiatric consultations from this study as anticipation of pain inhibits the coughing to clear airways of pulmonary secretions?

3.       As the study was conducted at two different times, there is a high degree of variability in postoperative pain among individuals. Does this affect consultations?

4.       Your team has mentioned various types of ambulatory surgery including orthopedics, plastic surgery, odontology, etc., however, there might be a varying degree of injury that may occur before or during the surgery and the recovery of the patient depending on age, gender, and other associated health conditions (Line no. 223-227, it is mentioned about the health status but no information about the sex/gender). Please elaborate on the measures you implemented to minimize the error.

5.       Conclusion part needs to be more elaborative. The authors seem to be rushing to prove the utility of Pharmacists over surgeons.

6.       The authors mentioned about a mean age of a total of 125 outpatients included is 45.7 years (SD 17.9). (Line no. 22-23) and then in line no. 168-169, they mentioned the mean age was 44.07 years (SD 17.38) in the control group and 47.28 years (SD 18.33) in the intervention group. Please explain.

Author Response

dear reviwer,

Thank you for the time you spent reviewing this manuscript and for your very constructive comments.

sincerely

the authors

Reviewer 2 Report

1)In the flow chart:

control group: what was the exclusion criteria?

interventional group: How is it possible to exclude 1717 from 1742 and still have 125?

2) the drop out rate did not allow a powered final sample. Please discuss. 

3) you report a difference in postoperative pain overall, but there is no difference when it comes to specific specialties. Also, the two groups are heterogenous in terms of surgical subspecialties and procedures. Please, elaborate. 

4) Patient satisfaction was not improved and there was no difference for severe pain. Please discuss. Would a better consultation with the anesthesiologist have a more favorable impact?

Author Response

Dear reviwer,

Thank you for the time you spent reviewing this manuscript and for your very constructive comments.

Sincerely

The authors

Reviewer 3 Report

Thank you for the opportunity to review this important manuscript. Here are my comments and suggestions.

Page 2, Line 71: there is a double blank space between the words ‘’ anesthesiologist’’ and ‘’and’’

In part ‘’2.4 Pharmacist consultation’’ describe in more detail how this consultation was done. Did the patients receive the phone numbers of the pharmacist? Were the patients physically introduced to the pharmacist before the operation? Was the pharmacist available 24 hours, 7 days a week or only available during the working hours of a pharmacist? Was the private phone number of a pharmacist given? Did the pharmacist know about all the conditions of the patients? In other words, did the pharmacist have electronic medical records of the patients? How did the pharmacist know which analgesia was prescribed? How did the pharmacist know the doses that the patient had taken?

Author Response

Dear reviewer

Thank you for the review, your relevant comment and your kindness. 

Sincerely

The authors

Reviewer 4 Report

Dear authors,

Thank you very much for providing this article on a very useful and important matter.

I’ve added some comments in order to improve the quality of your manuscript and make it more readable.

It is hoped that this will be constructive to you

Best regards

Reviewer

In the control group, 1552 out of 1677 patients were excluded the reason of which is uncertain. Please clarify this information.

In the intervention group, again there exists a very high rate of exclusion, eg. 1712 out of 1742 patients. A part of these patients were excluded probably because they have met the exclusion criteria. A part of data however, was not analyzed because of organizational shortcomings in providing pharmacist on ward. This data needs to be separated and explained accordingly.

Page 5. Line 181.

The authors talk sometimes about a control group and interventional group and thereafter talk about a retrospective and prospective group. It would be more sensible to use the same terminology throughout the entire text in order to reduce the misunderstanding.

This study compared two groups of outpatients over two different time periods with a large gap of one year in between. The surgeons needed to be present over two identical periods of the year.

This means technically a bias with respect to the medical practice, experience and performance of nurses, anesthesiologists, surgeons and the whole system on the awareness around the pain perception and treatment of patients. It is therefore necessary to acknowledge this and mention this great bias as one of the limitation.   

Page 7. Line 240

The authors assume that the community-hospital link could explain partly the decrease in pain perception in the intervention group. Could the authors provide some data in this regard such as how many patients at how many occasions did benefit from this advantage. Otherwise they should acknowledge this as a true assumption.

The discussion and conclusion sections need to be rewritten and corrected for English wording 

Last but not least, pain management is a multidisciplinary responsibility and should involve all team members including nurses, anesthesiologists, surgeons, both pharmacist in hospital and community, etc …. All these persons should be aware of that piece of pain management and should also get credit for their work. Adding pharmacist to your pain team is a worthy thing but the whole care system should be patient- and pain oriented.

Author Response

(The authors gave the same response as above.)

Round 2

Reviewer 3 Report

All queries answered.

Author Response

Dear reviewer,

Thank you for your help

Sincerely

the authors

Reviewer 4 Report

Dear authors, 

Thank you very much for this revised version. It's very convenient now. The text is very clear and understandable. 

I have no further suggestions or remarks to add but one. In table 3 where pain intensity in several groups has been displayed, please consider adding the pain scores and the threshold that are required to identify pain severity below the table. 

Wishing you all the best in the publication process

Best regards

Reviewer 

Author Response

Dear reviewer,

Once again, thank you for your careful review.

We have added the following statement under table 3. Please see the attachment

Sincerely

the authors
